# Yeast Translation Elongation Factor eIF5A Expression Is Regulated by Nutrient Availability through Different Signalling Pathways

**DOI:** 10.3390/ijms22010219

**Published:** 2020-12-28

**Authors:** Marina Barba-Aliaga, Carlos Villarroel-Vicente, Alice Stanciu, Alba Corman, María Teresa Martínez-Pastor, Paula Alepuz

**Affiliations:** 1Instituto Biotecmed, Facultad de Ciencias Biológicas, Universitat de València, C/Dr. Moliner 50, E46100 Burjassot, Spain; Marina.Barba@uv.es (M.B.-A.); carvivi@alumni.uv.es (C.V.-V.); stanciu@alumni.uv.es (A.S.); alba.corman@ki.se (A.C.); 2Departamento de Bioquímica y Biología Molecular, Facultad de Ciencias Biológicas, Universitat de València, C/Dr. Moliner 50, E46100 Burjassot, Spain; maria.teresa.martinez@uv.es

**Keywords:** eIF5A, mitochondrial respiration, gene expression, Hap1, heme, TOR, Snf1, iron

## Abstract

Translation elongation factor eIF5A binds to ribosomes to promote peptide bonds between problematic amino acids for the reaction like prolines. eIF5A is highly conserved and essential in eukaryotes, which usually contain two similar but differentially expressed paralogue genes. The human eIF5A-1 isoform is abundant and implicated in some cancer types; the eIF5A-2 isoform is absent in most cells but becomes overexpressed in many metastatic cancers. Several reports have connected eIF5A and mitochondria because it co-purifies with the organelle or its inhibition reduces respiration and mitochondrial enzyme levels. However, the mechanisms of eIF5A mitochondrial function, and whether eIF5A expression is regulated by the mitochondrial metabolism, are unknown. We analysed the expression of yeast eIF5A isoforms Tif51A and Tif51B under several metabolic conditions and in mutants. The depletion of Tif51A, but not Tif51B, compromised yeast growth under respiration and reduced oxygen consumption. Tif51A expression followed dual positive regulation: by high glucose through TORC1 signalling, like other translation factors, to promote growth and by low glucose or non-fermentative carbon sources through Snf1 and heme-dependent transcription factor Hap1 to promote respiration. Upon iron depletion, Tif51A was down-regulated and Tif51B up-regulated. Both were Hap1-dependent. Our results demonstrate eIF5A expression regulation by cellular metabolic status.

## 1. Introduction

Translation factor eIF5A is a small protein that is essential and highly conserved across eukaryotes, and with orthologues in prokaryotes and archaea. Interestingly, eIF5A is the only known protein to contain the amino acid hypusine, which is formed by the addition of a 4-aminobutyl group from polyamine spermidine to a specific conserved lysine residue. Posttranslational eIF5A hypusination occurs in two enzymatic steps: catalysed by deoxyhypusine synthase (DHPS) and deoxyhypusine hydroxylase (DOHH). DHPS and DOHH are also highly conserved and essential in most eukaryotes but are dedicated to modifying only one protein, which highlights the vital role of hypusinated eIF5A [1]. Though it was initially classified as a translation initiation factor, later studies revealed that eIF5A acts as an elongation factor that binds ribosomes at the E-site to project the hypusine-containing domain towards the P-site to promote the formation of peptide bonds between amino acid residues that are bad acceptors/donors for the reaction. The polypeptide motifs requiring eIF5A for their synthesis include stretches of consecutive prolines but also combinations of proline, glycine and charged amino acids ([2,3,4,5,6] and is reviewed in [7]).

Most eukaryotes, including human and yeast, have two genes that encode two extremely similar isoforms of eIF5A. Human genes *EIF5A-1* and *EIF5A-2* share 84% of amino acid sequence identity between the corresponding eIF5A encoded isoforms; yeasts *TIF51A* and *TIF51B* encode isoforms with 90% identity. Yeast and human eIF5A proteins also share more than 60% amino acid sequence identity and are functional homologues as heterologous human eIF5A expression allows yeast to grow with the deletion of eIF5A genes [8,9,10,11]. Despite producing very similar eIF5A proteins, the expression of eIF5A paralogue genes is dissimilar. In humans, only the eIF5A-1 isoform is abundant in most cell types, whereas eIF5A-2 expression is limited to the testis and brain [12]. However, both isoforms have been linked with different diseases in which they appear to be overexpressed. eIF5A-1 has been implicated in diabetes, several cancer types, viral infections and neurological diseases. On the contrary, eIF5A-2 is highly expressed in many cancers and its overexpression in certain cell types causes cellular transformation, for which it has been proposed to act as an oncogene [13,14,15,16].

In yeast, the genes encoding eIF5A isoforms *TIF51A* (also known as *HYP2*) and *TIF51B* (also known as *ANB1* and *HYP1*) are differentially expressed and reciprocally regulated by oxygen. Under aerobic conditions, *TIF51A* is highly and *TIF51B* is poorly expressed. *TIF51A* is essential, but *TIF51B* deletion has no effect on growth. On the contrary, *TIF51B* is up-regulated and *TIF51A* down-regulated when oxygen is lacking. *TIF51B* repression under aerobic conditions is triggered, similarly to other yeast genes induced by hypoxia, via the synergic action of DNA-binding proteins Rox1 and Mot3 through mechanisms that partially depend on the general repressor complex Ssn6/Tup1 [17,18,19,20]. The activation of repressor Rox1 in the presence of oxygen is produced via increased levels of heme groups, which are synthesised in mitochondria and serve as a secondary signal for oxygen. Then heme binds and activates nuclear transcriptional factor Hap1, which permits Hap1 to bind and promote the transcription of Rox1 and many genes required for oxygen utilisation and to control oxidative damage [21]. Conversely, in the absence of heme/oxygen, Hap1 becomes a repressor down-regulating *ROX1* and triggering the induction of *TIF51B* [22]. Unlike the known regulation of *TIF51B* by oxygen, knowledge about *TIF51A* regulation is scarce. It is supposed to be constitutively expressed under oxygen conditions, and *TIF51A* has been suggested to be positively regulated by Hap1 [17]. Additionally, the mechanism of *TIF51A* repression under anaerobiosis remains unknown.

Despite the two genes encoding eIF5A in both humans and yeast having clearly differential expression patterns, no evidence for a different molecular functionality of isoforms has been found. Indeed, in yeast cells, the expression of either eIF5A paralogue gene from a heterologous promoter restores yeast growth with the deletion of the *TIF51A* gene in rich media under aerobiosis conditions [8,10,11,23]. However, the differential expression of yeast *TIF51A* and *TIF51B* genes still suggests a functional specialisation of each eIF5A isoform, in which *TIF51A* would favour the metabolic adaptation to the presence of oxygen.

In the last few years, several reports have pointed out a function of eIF5A in the regulation of mitochondrial activity. Firstly, studies in mammals have reported the co-purification or localization of eIF5A with mitochondria [24,25]. Interestingly, the existence of an alternative human low expressed eIF5A-1 isoform, with an additional N-terminal extension containing a putative mitochondrial targeting sequence, has also been described. When overexpressed, this longer isoform co-purifies with mitochondria [26]. Secondly, there is evidence for the role of eIF5A in preserving mitochondrial morphology, distribution and integrity, obtained in *Schizosaccharomyces pombe* and mammals [27,28]. Thirdly, the connection between eIF5A and mitochondria-mediated apoptosis has been described, where eIF5A overexpression in human cells increased reactive oxygen species (ROS) and yielded loss of the mitochondrial transmembrane potential, and the release of cytochrome-c and caspase activation [29,30]. Fourthly, several reports in mammals have linked eIF5A with both cellular metabolism and respiration. In these studies, the inhibition of hypusinated eIF5A seems to reduce mitochondrial respiration, while leaving glycolytic ATP synthesis operative, which improves ischaemic conditions and organ transplantation [31,32,33]. Finally, and importantly, a recent report describes the modulation of mitochondrial respiration by eIF5A during macrophage activation [34]. Inhibiting the polyamine synthesis pathway and the hypusination of eIF5A blocks mitochondrial oxidative phosphorylation (OXPHOS). Moreover, hypusinated eIF5A is necessary to maintain the tricarboxylic acid (TCA) cycle and the electronic transport chain (ETC) integrity in macrophages by promoting the efficient translation of some mitochondrial enzymes with specific mitochondrial targeting sequences. Hypusinated eIF5A levels are modulated in response to immune stimuli, which suggests that hypusinated eIF5A activity is most critical in cells with increased respiration [34]. Briefly, increasing evidence links eIF5A and mitochondrial function, but whether the effect is direct or indirect, how eIF5A may affect the expression of mitochondrial proteins, and whether eIF5A expression is regulated by mitochondrial metabolism, and how, are questions that still have no answer.

Yeast cells are a feasible eukaryotic model to study the regulation of mitochondrial respiration as yeasts can grow aerobically or anaerobically, and the involved signalling pathways are well-known. In the presence of glucose and oxygen, yeast cells prefer fermentation to respiration because the former can proceed at much higher rates and allows more competitive growth and survival. Nevertheless, glucose fermentation to ethanol is energetically less efficient than aerobic respiration. Thus, under glucose and oxygen conditions, glycolysis and fermentation genes are induced, and key mitochondrial enzymes of the TCA cycle, ETC and OXPHOS, are subjected to glucose repression, as are other genes involved in the utilisation of other alternative carbon sources [35,36,37]. As yeast cultures in glucose media progress, sugar becomes limiting and yeast cells start metabolising the ethanol produced during fermentation by switching to aerobic respiration, which slows down growth. The transition from fermentation to respiration during this diauxic shift is produced by the up-regulation of TCA cycle, ETC and OXPHOS genes. This gene expression reprogramming involves different signalling pathways, such as protein kinase A (PKA), the target of rapamycin complex I (TORC1), Sch9, Snf1 and Mec1/Rad53, and requires several transcription factors, including Hap1, Hap2/3/4/5, Cat8 and Rgt1/3, as well as communication between the nuclear and mitochondrial genome, and crosstalk between pathways [35,36,37,38,39,40,41].

Under high glucose conditions, glucose repression is executed mainly by transcription factor Mig1, together with the Ssn6/Tup1 complex. Snf1, the yeast homologous to mammalian AMP-activated kinase, is the master kinase that removes glucose repression. Snf1 is inhibited by glucose and stimulated when glucose is limiting. Activated Snf1 inhibits Mig1-mediated repression by its phosphorylation and subsequent translocation to the cytoplasm [35,36,37,42]. High glucose maintains the PKA and TORC1 signalling pathways active and promotes proliferation by inducing the expression of the ribosome and translation machinery genes, and by inhibiting mitochondrial respiration. Thus, the activity of PKA and TORC1 acts conversely to that of Snf1. Indeed, negative reciprocal regulation between PKA and Snf1, and between the TORC1 and Snf1 signalling pathways, has been documented [43,44,45,46,47].

Under glucose depletion, or with non-fermentative carbon sources, the expression of the TCA cycle, ETC and OXPHOS genes requires transcription complex Hap2/3/4/5 [35,48,49]. HAP complex activity depends on heme cellular levels and, independently, on PKA and Snf1, which suggests that separate pathways can control mitochondrial respiration [21,50]. Activation subunit Hap4 is the only one of the HAP complexes to be up-regulated upon glucose depletion [51]. Part of this Hap4 up-regulation is mediated by transcription factor Cat8, which is induced and activated by Snf1 [35,38]. It is noteworthy that lowering glucose levels in aerobic growing yeast increase the amounts of pyruvate directed to mitochondria which, in turn, provides substrates for heme biosynthesis. Consequently, the increased heme cellular level stimulates Hap4 transcriptional activity in a Hap1-dependent manner and independently of Snf1. These results suggest that the glucose repression of respiration is partly due to the low metabolic flux in the TCA cycle and, therefore, to the low cellular heme level under high glucose [52].

To deepen in the putative role of eIF5A in mitochondrial function, we studied the need for eIF5A to metabolise non-fermentative carbon sources through respiration and eIF5A expression regulation under fermentative and respiratory conditions. We herein document that isoform Tif51A, but not Tif51B, is required for growth in media with glycerol and ethanol as carbon sources. Under these conditions, *TIF51A* was up-regulated in a Hap1- and Snf1-dependent manner, and the depletion of *TIF51A* reduced oxygen consumption, but full eIF5A hypusination was not required to maintain high respiration levels. Our results suggest that during the diauxic shift, *TIF51A* is initially repressed by reduced TORC1 activity, but later it is Hap1-induced due to the increase in the metabolic flux in the TCA cycle and, consequently, in heme cellular levels. Altogether, our results suggest that the Tif51A isoform of yeast eIF5A responds to the metabolic state of cells to promote mitochondrial respiration.

## 2. Results

### 2.1. The Tif51A Isoform of Yeast eIF5A Is Required for Respiration and Growth with Non-Fermentative Carbon Sources

*S. cerevisiae* preferentially ferments glucose, even in the presence of oxygen but, upon glucose deprivation and during growth with non-fermentable carbon sources, many respiratory genes are derepressed and highly induced for energy production [35,36,37]. As different studies suggest a role of eIF5A in mitochondrial functioning and based on the recently described participation of hypusination and polyamines in modulating the expression of the mitochondrial proteins involved in respiration [34], we were interested in investigating whether eIF5A is actually needed for yeast respiration. To do so, we carried out some experiments on non-fermentable substrates, such as glycerol and ethanol, or a mainly respiratory substrate as galactose, to show different degrees of respiratory rates [39]. Under these conditions, mitochondrial oxidative phosphorylation processes are mandatory for cell growth and proliferation. At semi-restrictive temperatures, growth defects of the temperature-sensitive *tif51A-1* mutant, carrying a single (Pro83 to Ser) mutation in the yeast Tif51A isoform of eIF5A, and more severe defects of the *tif51A-3* mutant, carrying a double (Cys39 to Tyr, Gly118 to Asp) mutation, but not of strain *tif51B*∆, were observed (Figure 1a). Next, we assessed the expression of the two eIF5A isoforms upon growth under non-fermentative conditions, and we observed regulation in opposite ways. While the mRNA levels of *TIF51A* significantly increased, *TIF51B* levels lowered compared to glucose (Figure 1b).

In order to more directly explore the requirement of eIF5A in mitochondrial respiration, we measured the oxygen consumption rate in both wild-type (WT) and eIF5A temperature-sensitive mutant cells. The relative oxygen consumption in both mutant cells at a non-permissive temperature was significantly reduced compared to the WT strain, especially in mutant *tif51A-3* (Figure 1c). Minor differences were also observed at the permissive temperature between the wild type and mutants in line with the expected slight loss of function of these Tif51A mutations [4,53]. To take advantage of the fact that the last eIF5A hypusination step is not essential in yeast cells [54], we investigated the implication of full hypusination in the requirement of eIF5A for cell respiration. However, no differences in the oxygen consumption rate were observed between the wild type and the DOHH (*LIA1*) mutant cells (Figure 1e). Collectively, these results indicate that mitochondrial respiration is compromised when the yeast Tif51A isoform of eIF5A is lacking, regardless of its hypusination state.

### 2.2. Snf1 and Hap1 Signalling Pathways Are Involved in TIF51A Induction under Respiratory Conditions

As previously described, *TIF51A* expression was induced in the cells grown with non-fermentable carbon sources glycerol and ethanol, which are used for the oxidative metabolism of mitochondria. We aimed to investigate the mechanism exerted to achieve distinct modulations in eIF5A abundance. PKA, Snf1 and the heme responsive Hap1 and Hap2/3/4 complex are the main factors involved in the metabolic reprograming between two alternative physiological states: fermentation and respiration. PKA affects gene expression, mostly by proteins Msn2 and Msn4, known as general stress-responsive transcription factors. The activators Msn2/Msn4 are down-regulated by PKA under high glucose, and up-regulated by Snf1 under respiratory conditions, which contribute to the adaptive response to respiration [55,56,57]. We, hence, used mutants *hap1*∆, *snf1*∆ and *msn2*∆, which grew well in 2% glucose media and still grew in the media with a non-fermentative carbon source, although *snf1*∆ displayed a major growth defect (Figure 2a). However, we were unable to work with the *hap4*∆ mutant in the media with ethanol or glycerol as the deletion of any Hap complex subunit did not enable cells to grow under these conditions (Figure 2a and [48]). We determined the mRNA levels of *TIF51A* when cells were grown under non-fermentable substrates. Mutants *hap1*∆ and *snf1*∆ showed no increase in the *TIF51A* levels compared to the WT or *msn2*∆ (Figure 2b). We established the expression of the flavoprotein subunit of the mitochondrial enzyme succinate dehydrogenase (SDH1) as an additional control, which is induced under respiratory conditions. We found that the mRNA levels were affected only in *hap1*∆ and *snf1*∆ (Figure 2c). These results indicate that in the cells grown in the non-fermentable carbon sources, the Hap1p and Snf1p pathways play a role in the up-regulation of eIF5A expression. Based on these results, we hypothesised that Hap1 could influence respiration through eIF5A activation, among its other already known target genes.

### 2.3. TIF51A Expression Drops during the Diauxic Shift But Subsequently Increases in a Hap1-Dependent Manner

We aimed to investigate whether the up-regulation of *TIF51A* expression under non-fermentative carbon sources would also be reflected by up-regulation when the cells grown in glucose media underwent a metabolic shift from fermentation to respiration, and whether this regulation would also be mediated by Hap1. To test this, we measured and compared the mRNA levels of the two eIF5A isoforms in the WT and *hap1*∆ mutant cells in the exponential and post-diauxic growth phases in a YPD batch culture for up to 4 days before reaching the stationary phase and entering the quiescent state (Figure 3a). Figure 3b shows how the mRNA levels of *TIF51A* significantly increased to almost 2-fold in the WT after 48 h vs. the expression under exponential growth (time 0), but remained constant, or even slightly lowered, in *hap1*∆. However, the *TIF51B* levels significantly and continuously decreased (Appendix A). To compare the regulation of *TIF51A* with that of another translation factor, we studied translation initiation factor eIF2A expression levels. In the WT strain, with no differences in the *hap1*∆ strain, the *eIF2A* mRNA levels slowly lowered from 24 h, which continued up to 96 h of incubation in YPD (Figure 3c). Our result with eIF2A agrees with previous results, which show that the expression of most translation factors decreases as cells enter the post-diauxic phase and face lack of glucose [51]. To confirm the transition to a respiratory metabolism, we measured TCA enzyme subunit *SDH1* expression and observed a substantial increase after the metabolic switch in both strains: WT and *hap1*∆ (Figure 3d). An increase in genes TCA, ETC and OXPHOS under respiratory conditions occurs mainly under the control of Hap2/3/4, and Hap1 activity is also required for some genes [21,35,48,49,58]. The up-regulation of Hap4 upon glucose depletion has also been documented [51]. In our experiment, we observed a marked increase in the *HAP4* mRNA levels at 24 h of culture in the WT cells (Figure 3e). Interestingly, *HAP1* also showed a 5-fold increase at 24 h and remained significantly up-regulated for longer times (Figure 3f). Finally, we also checked any variation in the eIF5A protein levels, although using the anti-eIF5A antibody cannot discriminate between isoforms Tif51A and Tif51B, and the eIF2A protein as the control. As shown in Figure 3g,h, the eIF5A protein level dropped to almost undetectable levels after 24 h of growth in YPD, which corresponds to the diauxic shift. This level returned to the initial levels at 72 h. This result may reflect, on the one hand, the observed reduction in *TIF51B* expression but, given the much higher *TIF51A* expression under the basal conditions [10], and as *TIF51A* mRNA does not drop at 24 h, this scenario suggests the down-regulation of Tif51A at the translation or protein stability levels. On the other hand, the later recovery of the Tif51A protein level may result from the increase in its mRNA levels after 48 h (Figure 3g,h).

Altogether, these results point out the specific requirement of Hap1 for inducing *TIF51A* after a shift from fermentation to respiratory growth, but also indicate that eIF5A regulation clearly differs from other translation factors. While the expression of most translation factors, if not all, decreased after the metabolic shift, Tif51A showed a very different and, possibly dual, regulation. On the one hand, its expression drastically dropped immediately after the diauxic change, which suggests a rapid response to lowering glucose levels. On the other hand, this decrease was followed by a progressive increase in expression, which may imply that this eIF5A isoform is likely to play a role in the respiratory process.

### 2.4. Glucose Availability and TORC1 Regulate eIF5A Expression

In an effort to understand eIF5A regulation at the YPD incubation times when the glucose level drops, we decided to study the conceivable TORC1-mediated regulation. Like most organisms, yeast coordinates protein biosynthetic capacity to nutrient availability through the TORC1 signal transduction pathway. Under unfavourable growth conditions, TORC1 is inactive, which leads to a slow reduction in translation and synthesis of ribosomal components [59,60,61].

We confirmed that TORC1 signalling inhibition by rapamycin treatment led to the rapid and pronounced down-regulation of the mRNA levels of the two eIF5A isoforms (*TIF51A* and *TIF51B*) and of that of eIF2A (Figure 4a). Likewise, upon TORC1 deactivation, the protein levels of both eIF5A and eIF2A translation factors significantly lowered (Figure 4b). To further prove the regulation upon glucose availability, we studied the mRNA levels of *TIF51A* in three different scenarios (described in the Materials and Methods section): (1) glucose concentration drops from 2 to 0.1%; (2) glucose concentration rises from 0.1 to 2%; (3) the same increase in glucose concentration as (2) but supplemented with rapamycin. The results showed that the *TIF51A* levels lowered after a drop in glucose concentration to 0.1%, but were rescued when glucose was added back to cells at regular levels (Figure 4c,d). Moreover, rapamycin treatment, even with excess glucose, did not rescue the higher levels reached in scenario 2, which implies a TORC1-mediated response to glucose availability (Figure 4e). No big differences were observed between the *TIF51A* and *eIF2A* mRNA levels in the three scenarios, which means that both translation factors are regulated in the same way upon changes in nutrient accessibility.

### 2.5. An Increase in the Metabolic Flux in the TCA Cycle and at Heme Cellular Levels Up-Regulates TIF51A in a Hap1-Dependent Manner

The respiratory process is controlled by the carbon source, together with oxygen and heme levels. The above-stated experiments indicate *TIF51A* regulation under respiratory conditions by Hap1, which is consistent with previous observations [17]. Hap1 is known to respond to both heme and non-fermentable energy. As Zhang et al. described [52], marked pyruvate transport into mitochondria results in high heme levels and, thus, enhanced Hap1 and Hap2/3/4/5 complex activities. Heme synthesis starts in mitochondria and is limited by TCA and succinyl-CoA availability. To better understand the possible regulation of eIF5A by Hap1, we investigated whether the heme levels and flux into the TCA cycle are critical for *TIF51A* transcriptional regulation.

We examined the *TIF51A* expression in the WT, *hap1*∆, *mpc1*∆ and *pda1*∆ cells. Pda1 is a subunit of the pyruvate dehydrogenase complex which catalyses the conversion of pyruvate into acetyl-CoA in mitochondria, while Mpc1 is a pyruvate transporter localised in the inner mitochondrial membrane. We observed that both the *TIF51A* and *SDH1* expression levels in galactose medium significantly lowered in *hap1*∆, *mpc1*∆ and *pda1*∆ compared to the WT (Figure 5a,b). We interpret these results to mean that the metabolic flux into the TCA cycle regulates eIF5A expression, most likely by regulating Hap1 expression.

As Zhang et al. stated [52], the effect of increasing heme levels on HAP transcription appears to be more significant in those cells grown in glucose than in galactose. To determine whether the heme level is limiting for *TIF51A* transcription in WT cells, we added 5-aminovulenic acid (ALA, the second metabolite of the heme biosynthesis pathway) or hemin (heme derivative) to the cells grown in the rich media containing glucose. We found that the addition of extracellular ALA or hemin increased *TIF51A* expression almost 2-fold, and a similar increase was observed for *HAP1* and *SDH1* expression (Figure 5c–e). Altogether, these results suggest that *TIF51A* transcription is regulated by Hap1 which, in turn, is regulated by heme and by the metabolic flux into the TCA cycle. This could also explain the increase in *TIF51A* expression that took place during the diauxic shift (Figure 3b) when the metabolic flux into mitochondria increased upon glucose exhaustion.

### 2.6. TIF51A Is Regulated under Iron Deficiency in a Hap1-Dependent Manner

Respiration is a highly iron-consuming process as both the TCA cycle and ETC require iron and heme in many steps. Indeed, during iron starvation, cells are unable to grow under non-fermentable carbon sources. Iron deficiency regulation in *S. cerevisiae* involves metabolic remodelling, which is achieved by changes in gene expression at the transcriptional and post-transcriptional levels to prioritise iron-dependent essential cellular processes over non-essential processes, including mitochondrial respiration (reviewed in [62]). According to our previous results, we hypothesised that under iron starvation, respiratory process inhibition would also involve the down-regulation of *TIF51A* expression.

The mRNA levels of the two eIF5A isoforms were determined under iron deficiency to test the possible iron-dependent activity of eIF5A. Iron starvation significantly affected the expression of the two isoforms. While *TIF51A* expression decreased in a time-dependent manner, *TIF51B* expression increased almost 4-fold (Figure 6a). Due to the very low basal *TIF51B* expression, its increase did not compensate the down-regulation of *TIF51A*, and eIF5A protein levels (corresponding to both eIF5A isoforms recognized by the antibody) decreased under iron depletion (Appendix A). The increased but still low expression of the Tif51B isoform must be insufficient to support yeast growth because we observed that the complete depletion of the Tif51A protein (using *TIF51A* temperature sensitive mutants at restrictive temperature) rendered it sensitive to iron starvation (Appendix A). Next, we determined if eIF5A regulation under iron deficiency was based on a transcriptional mechanism and attempted to identify the implicated factor. To that end, we tested the expression of both isoforms under iron sufficiency or deficiency conditions in the WT, *hap1*∆ and *hap4*∆ mutant cells. We found that only for *hap1*∆ cells the *TIF51A* mRNA levels remained unchanged and transcriptional regulation was lost, which indicate that Hap1 is responsible for eIF5A regulation under iron deficiency (Figure 6b). The synthesis of the iron-containing heme directly correlates with iron availability, and it has been shown that, under iron-limiting conditions, the cytochrome c-encoding *CYC1* gene is transcriptionally down-regulated via the Hap1 transcription factor with some contribution made by the Hap2/3/4/5 complex [63]. We observed that the *CYC1* mRNA levels were down-regulated and almost undetectable when iron was absent (Figure 6c). With the case of *hap1*∆ and *hap4*∆ cells, the *CYC1* levels substantially lowered under the iron-sufficient condition and most of the down-regulation was lost in the single mutants, which agrees with previous data [63]. Finally, Hap1 was also important for *TIF51B* regulation as the induction that occurred in the WT strain under iron depletion was blocked when cells were Hap1-deficient (Figure 6d). This result can be interpreted by the fact that lack of heme, in this case caused by limited iron availability, converts Hap1 in a repressor that down-regulates ROX1, which is necessary for inducing *TIF51B* [22]. Taken together, these results reinforce the idea of *TIF51A* expression regulation at the transcriptional level by Hap1 to up- or down-regulate its expression depending on the cell’s metabolic requirements.

## 3. Discussions

The adaptation of the cellular metabolism to external circumstances is important for those unicellular systems that must deal with a continuously changing environment, but also for multicellular ones as redirecting metabolism can promote different cellular functions. One important example of metabolic adaptation is the well-known Warburg effect, by means of which most tumour cells sustain aerobic glycolysis with glucose fermentation into lactate, unlike complete glucose oxidation by mitochondria, to meet their bioenergetic and anabolic demands [64,65]. In fact, it has been proposed that all high proliferating cells adapt their metabolism to facilitate the uptake and incorporation of nutrients into the biomass needed to produce a new cell (aerobic glycolysis) over the promotion of high-efficient ATP synthesis in the quiescent (differentiated) state (mitochondrial OXPHOS). This also applies to microorganisms like yeast cells, which prefer the fermentation of glucose when it abounds, but change to mitochondrial respiration when glucose is scarce [66]. Understanding how cells adapt their metabolism to meet demands is relevant because wrong adaptation can have pathological consequences. Previous studies (detailed in the Introduction) as well as the present one suggest a role of eIF5A in promoting mitochondrial metabolism.

Our study addresses how eIF5A expression is regulated to adapt it to metabolic requirements. Although eIF5A is a highly expressed protein and has been described as one of the 20 most abundant proteins in proliferating cells [67], 2- to 4-fold increases in yeast cells took place under respiratory conditions. Like other proteins involved in translation [59,60,61], we document a positive regulation by the TORC1 signalling pathway activated under abundant nutrient conditions. In this situation, we expected the main function of cellular eIF5A to facilitate the cytoplasmic translation of the genes encoding proteins with specific amino acids motifs [2,3], although other molecular functions for eIF5A have been described [68]. With low glucose or non-fermentative carbon sources, our results indicate that the Tif51A isoform of yeast eIF5A is up-regulated by Snf1 and Hap1, and Tif51A depletion compromises growth in ethanol or glycerol and reduces oxygen consumption. The reason why more eIF5A protein is needed under this condition when growth has slowed down and, therefore, less cytoplasmic translation is required, is not known. A model summarising our results is show in Figure 7. It is worth highlighting the recent results reported by Puleston et al. [34], who suggest that eIF5A is required for the efficient translation of a subset of mitochondrial proteins, such as succinate dehydrogenase (SDH) and some other TCA cycle and OXPHOS proteins, which carry specific mitochondrial target signals. On the contrary, other TCA enzymes would be less eIF5A-dependent, which eIF5A inactivation would not affect their expression. A search for eIF5A-dependent tripeptide motifs [2] reveals a slight enrichment of these motifs in TCA enzymes but an underrepresentation of these motifs in OXPHOS proteins, both compared to total yeast proteins (Appendix A). Interestingly, polyproline motifs are excluded in the polypeptide sequences of both functional groups (Appendix A). A more exhaustive investigation will be necessary to identify eIF5A mitochondrial targets and mechanisms involved in the eIF5A control of mitochondrial respiration. The need of eIF5A for synthesizing specific mitochondrial proteins would explain its upregulation under respiratory conditions, when the expression of some of the proteins required for respiration substantially increases (*SDH1* mRNA increases more than 100-fold during the diauxic shift, Figure 3d). If this role of eIF5A in the translation of nuclear-encoded mitochondrial proteins is direct, the co-purification of eIF5A with mitochondria [24,25,26] may indicate that eIF5A is involved in the described translation on the yeast mitochondrial surface, where inner-membrane proteins like SDH are supposed to be cotranslationally targeted to the mitochondria [69].

One outcome of our study was the clearly differential regulation of the yeast eIF5A isoforms. Their opposite regulation by oxygen has been previously documented, with *TIF51A* being repressed and *TIF51B* activated during hypoxia/anaerobiosis, when heme-dependent Hap1 activity controls *TIF51B* repressor Rox1 expression [17,18,19,20,21,22]. Here we show that Hap1 is also the main factor that controls *TIF51A* expression in response to respiratory conditions, heme levels and iron availability. In fact, a putative binding site for Hap1 (CGGnnnTAnCGG, [70]) exists 571 bp upstream of the ATG of *TIF51A*, but further work should be done to demonstrate the direct regulation of *TIF51A* expression by Hap1. The control of eIF5A isoforms by Hap1, likely by activating *TIF51A* expression and the Rox1 repression of *TIF51B* under nutrient and oxygen availability, but by repressing *TIF51A* and Rox1 under nutrient and/or oxygen scarcity, would allow the opposite regulation of the two paralogue genes with only one factor. It is tempting to suggest that the differential expression of the eIF5A isoforms would promote, in each case, differential metabolic outcomes with Tif51A promoting respiration and Tif51B promoting aerobic glycolysis. To date, there has been no evidence to support this in yeast, although it has been described that the eIF5A-2 isoform of human eIF5A promotes aerobic glycolysis in human hepatocellular carcinoma (HCC) [71]. Thus eIF5A-2 overexpression increases growth, glucose uptake and lactate secretion by up-regulating glycolytic enzymes [71], which is precisely the metabolic reprogramming that occurs in most cancer cells [72].

Several aspects of *TIF51A* regulation by the metabolic status of cells will have to be explored in future studies. We showed that the up-regulation of *TIF51A* under non-fermentative carbon sources also depends on Snf1 kinase. However, we do not know if this regulation is executed by the release of Mig1-repression or through transcription activators under the control of Snf1, such as Cat8, which controls the derepression of *HAP4* under respiratory conditions [35]. The control of *TIF51A* expression by glucose and TORC1 should also be further investigated to elucidate the involved mechanism, which could implicate TORC1 downstream kinase Sch9 because its depletion increases respiration [43]. One of our observations indicates that the deletion mutant in yeast DOHH enzyme *lia1*∆ has the same oxygen rate consumption as the WT strain. Differently to mammalians and most eukaryotes, *S. cerevisiae* DOHH is not essential and shows similar growth to the WT [54]. Our results suggest that full yeast Tif51A hypusination would not be necessary to promote respiration, conversely to the results obtained for mammals [34].

The control by Hap1 and Snf1 of *TIF51A* expression under respiratory conditions that we herein report is likely to occur at the transcriptional level. However, it has also been previously reported that the posttranscriptional control of *TIF51A* expression by carbon sources is executed at the mRNA stability level. *TIF51A* mRNA harbours an ARE (AU-rich) element) at the 3′ UTR (untranslated region), which has been described to stabilise the mRNA on glucose and destabilise it in glycerol media [73]. Although this regulation is opposite to what we show here, with a higher *TIF51A* mRNA level in glycerol than in glucose, both results may still be compatible if major differences at the transcriptional level would reverse the negative impact of a reduction in *TIF51A* mRNA stability in glycerol media. AREs are also important for mRNA stability regulation and translation upon iron depletion through mRNA binding proteins Cth2 and Cth4. Iron deficiency promotes the specific binding of Cth2/4 to ARE-containing mRNAs and down-regulates, among other non-essential processes, mitochondrial respiration [74,75,76]. Whether *TIF51A* mRNA is posttranscriptionally regulated through its ARE sequence must be addressed in future studies.

## 4. Materials and Methods

### 4.1. Yeast Strains and Growth Conditions

All the *Saccharomyces cerevisiae* strains used herein are listed in Appendix A. Yeast cells were grown in liquid YPD (2% glucose, 2% peptone, 1% yeast extract), YPGal (2% galactose, 2% peptone, 1% yeast extract), YPGly (2% glycerol, 2% peptone, 1% yeast extract), YPEtOH (2% ethanol, 2% peptone, 1% yeast extract) or synthetic complete (SC) media.

A PCR-based genomic disruption technique was employed to replace genomic full length *HAP1* ORF with the KanMX marker. Plasmid pFA6a-KanMX6 [77] was used as a template for PCR reactions using primers HAP1-F1 and HAP1-R1 for *HAP1* deletion. The resulting cassette was transformed into the WT strain by the lithium acetate-based method [78] and transformants were selected in the YPD medium supplemented with geneticin (G418, Gibco Life Technologies, Waltham, MA, USA).

Experimental assays were performed with the cells exponentially grown for at least four generations at 30 °C until required OD_600_. Temperature-sensitive strains were grown at the permissive temperature of 25 °C until required OD_600_ and transferred to the non-permissive temperature of 37 °C for 4 h for complete eIF5A depletion. For some experiments described in the text, the media were supplemented with 100 µM of bathophenanthrolinedisulphonic acid (BPS, Sigma, St. Louis, MO, USA), 200 ng/mL of rapamycin (LC Laboratories, Woburn, MA, USA), 300 µg/mL of δ-aminolevulinate (ALA, Sigma) and 25 µg/mL of hemin (Sigma). ALA and hemin were dissolved in water and DMSO at 100 mg/mL and 10 mg/mL, respectively, and added to media at the indicated concentrations.

### 4.2. Western Blot Analysis

Protein extraction and Western blot analyses were performed as previously described [79]. In brief, a cell culture volume corresponding to 5–10 OD_600_ units was harvested by centrifugation. For protein extraction, cell pellets were washed and resuspended in 200 µL of NaOH 0.2M and incubated at room temperature for 5 min for posterior centrifugation at 12,000 rpm for 1 min. Then samples were resuspended in 100 µL of 2X-SDS protein loading buffer (Tris-HCl pH 6.8 24 mM, Glycerol 10%, SDS 0.8%, β-mercaptoethanol 5.76 mM, bromophenol blue 0.04%) and boiled for 5 min at 95 °C. Afterwards, lysates were centrifuged at 3000 rpm for 10 min at 4 °C to remove cell debris and insoluble proteins, and supernatants were transferred to fresh tubes and stored at −20 °C. The soluble protein content in the extract was quantified by an OD_280_ estimation in a Nanodrop device (Thermo Fisher Scientific, Waltham, MA, USA) to load equal protein amounts per sample into 15% SDS-PAGE gels.

SDS-PAGE and Western blot were performed by standard procedures (Bio-Rad Laboratories). Membranes were blocked with 5% skimmed milk in TBS-T (150 mM NaCl, 20 mM Tris, 0.1% Tween20, pH 7.6) for 1 h at room temperature and incubated with primary antibodies overnight at 4 °C against eIF5A (rabbit polyclonal 1:500, Abcam ab137561), hypusinated-eIF5A (FabHpu antibody, 1:600, Genentech, San Francisco, CA, USA), eIF2A (rabbit polyclonal 1:1000, kindly provided by T. Dever) or glyceraldehyde-6-phosphate dehydrogenase (rabbit polyclonal anti-G6PDH antibody, 1:20,000, Sigma A9521). Bound antibodies were detected using the appropriate horseradish peroxidase-conjugated secondary antibodies (1:10,000 Promega). Chemiluminiscent signals were detected with an ECL Prime Western blotting detection kit (GE Healthcare, Chicago, IL, USA) and digitally analysed by the ImageQuant LAS 4000 software (GE Healthcare). Band intensity was normalised against G6PDH bands. At least three replicates of each sample were analysed.

### 4.3. RT-qPCR Analysis

For the analysis of mRNA levels, total RNA was isolated from yeast cells following the phenol:chloroform protocol. Briefly, a volume of an exponential phase culture corresponding to 10 OD_600_ units was harvested and flash frozen. The cells were resuspended in 500 μL of cold LETS buffer (LiCl 0.1 M, EDTA pH 8.0 10 mM, Tris-HCl pH 7.4 10 mM, SDS 0.2%) and transferred to a screw-cap tube already containing 500 μL of sterile glass beads and 500 μL of phenol:chloroform (5:1). Then, the cells were broken in a Precellys 24 tissue homogenizer (Bertin Technologies) and centrifuged. The supernatant was transferred o a new tube containing 500 μL of phenol:chloroform (5:1) and then to a tube containing 500 μL of chloroform:isoamyl alcohol (25:1). The RNA from the top phase was precipitated and finally dissolved in water for later quantification and quality control using a Nanodrop device (Thermo Fisher Scientific).

The reverse transcription and quantitative PCR reactions were performed as detailed in [80]. Briefly, 2.5 µg of the total DNAse-I-(Roche) treated RNA were retrotranscribed using an oligo d(T)18 with Maxima Reverse Transcriptase (Thermo Fisher Scientific). cDNA was labelled with SYBR Pre-mix Ex Taq (Tli RNase H Plus, from Takara) and Cq values were obtained from the CFX96 TouchTM Real-Time PCR Detection System (BioRad). Endogenous ACT mRNA levels were used for normalisation. At least three biological replicates of each sample were analysed, and the specific primers designed to amplify the gene fragments of interest are listed in Appendix A.

### 4.4. Oxygen Consumption Assays

Molecular oxygen consumption was measured by a model Oxyview 1 System (Hansatech) and an S1 Clark-type oxygen electrode following the manufacturer’s protocol. Cells were grown to an OD_600_ of 1.5–2 in YPGal medium, and a volume corresponding to 1.0 OD_600_ was collected and washed with distilled water. The cells were then resuspended in 1 mL of the YEP medium containing 2% ethanol and 3% glycerol, and were transferred to the oxygen consumption chamber, magnetically stirred and maintained at 30 °C. The oxygen content decline was monitored for 15 min and respiratory rates were determined from the slope. The oxygen consumption rate in the WT cells was set at 1.

## Figures and Tables

**Figure 1 ijms-22-00219-f001:**
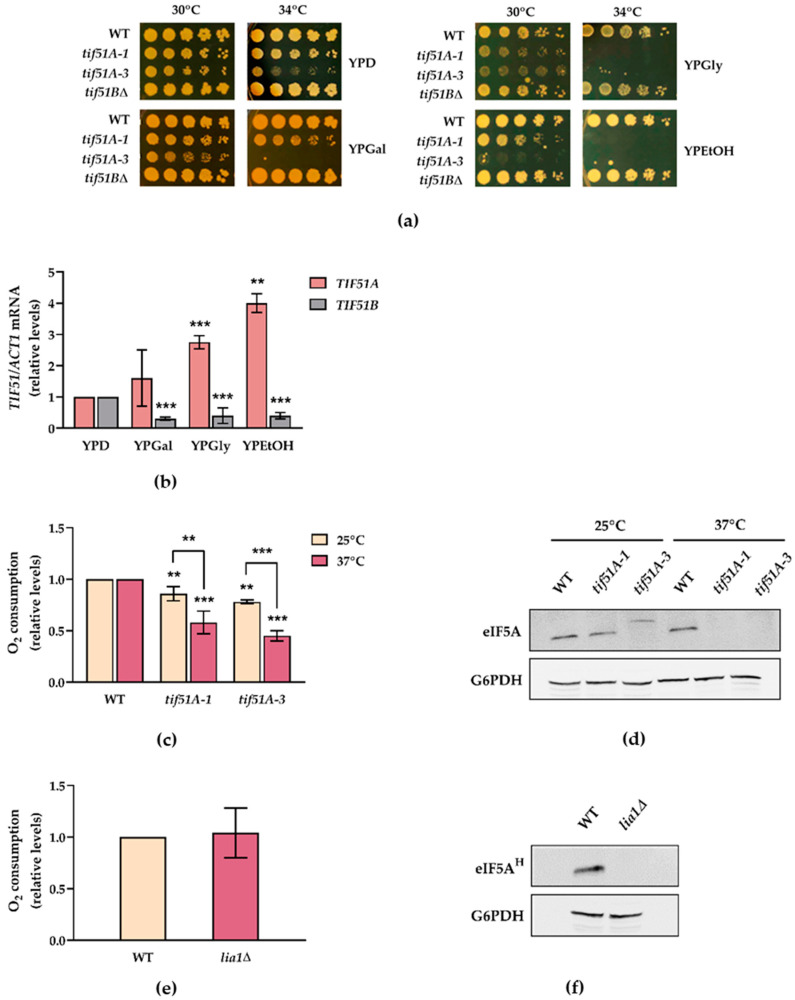
*TIF51A* is required for respiration independently of hypusination. (**a**) Growth of the WT, *tif51A-1*, *tif51A-3* and *tif51B*∆ strains was tested in YEP medium containing 2% glucose (YPD), 2% galactose (YPGal), 2% glycerol (YPGly) or 2% ethanol (YPEtOH) at the indicated temperatures. (**b**) The WT cells were grown in YPD, YPGal, YPGly or YPEtOH to the exponential phase (OD_600_ of 0.5). Relative *TIF51A* and *TIF51B* mRNA levels were determined. (**c**) The WT, *tif51A-1* and *tif51A-3* strains were grown in YPGal at permissive temperature (25 °C) and transferred to non-permissive temperature (37 °C) for 4 h until an OD_600_ of 1.5–2 was reached. Relative oxygen consumption rates are shown. (**d**) Western blotting of eIF5A in the WT, *tif51A-1* and *tif51A-3* cells at 25 °C and 37 °C showing eIF5A depletion. Glucose-6-phosphate dehydrogenase (G6PDH) protein levels were used as loading controls. *tif51A-3* mutated protein migrates slightly more slowly than the wild-type Tif51A and *tif51A-1* mutant due to the mutation of glycine 118 to aspartic. (**e**) The WT and *lia1*∆ strains were grown in YPGal at 30 °C until an OD_600_ of 1.5–2 was reached. Relative oxygen consumption rates are shown. (**f**) Western blotting of hypusinated eIF5A in the WT and *lia1*∆ showing differences in hypusination. G6PDH protein levels were used as the loading controls. (**b**,**c**,**e**) The results are shown as the means ± SD of three independent experiments and expressed in relation to the value for the 2% glucose condition or the WT. Statistical significance was measured by a Student’s *t*-test in relation to the 2% glucose condition or WT. ** *p* < 0.01, *** *p* < 0.001.

**Figure 2 ijms-22-00219-f002:**
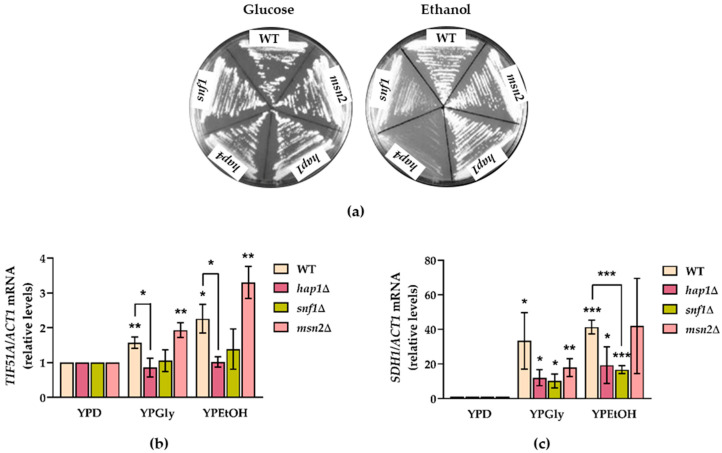
The Hap1 and Snf1 signalling pathways are required for *TIF51A* induction under respiratory conditions. (**a**) Growth of the WT, *hap1*∆, *snf1*∆, *msn2*∆ and *hap4*∆ strains on YEP medium plates containing 2% glucose or 2% ethanol. (**b**,**c**) The WT, *hap1*∆, *snf1*∆ and *msn2*∆ strains were grown in YEP medium containing 2% glucose, 2% glycerol or 2% ethanol to the exponential phase (OD_600_ of 0.5). Relative *TIF51A* (**b**) and *SDH1* (**c**) mRNA levels were determined. The results are shown as the means ± SD of three independent experiments and expressed in relation to the value for the 2% glucose condition. Statistical significance was measured by a Student’s *t*-test in relation to the 2% glucose condition or WT. * *p* < 0.05, ** *p* < 0.01, *** *p* < 0.001.

**Figure 3 ijms-22-00219-f003:**
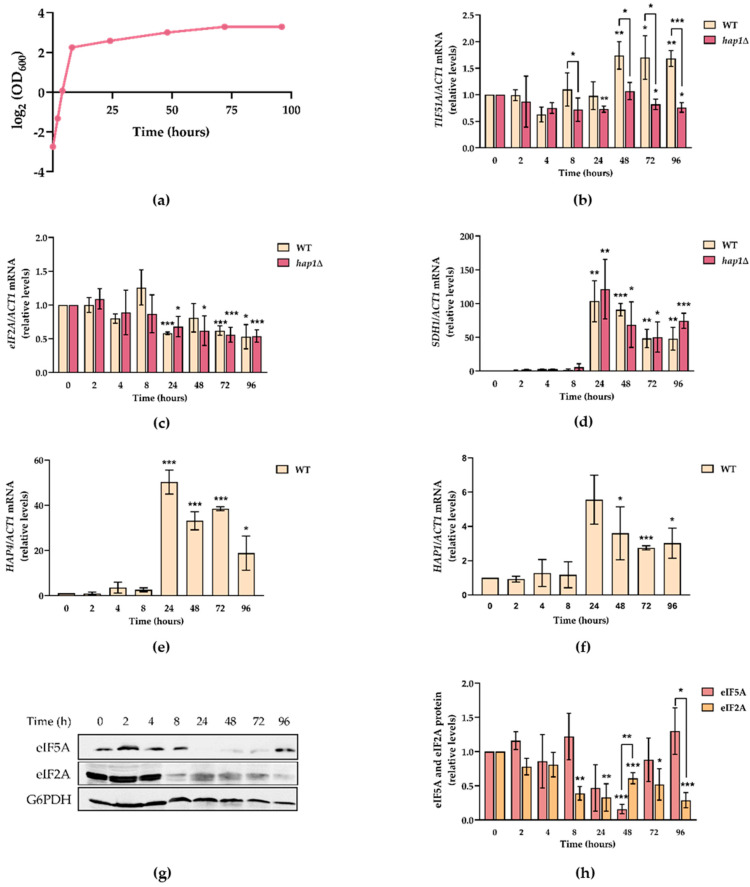
Tif51A expression drops during glucose exhaustion but recovers in a Hap1-dependent manner in the post-diauxic phase. (**a**) The WT *and hap1*∆ cells were grown in YPD medium for 96 h and samples were collected at the indicated time points. (**b**–**f**). Relative *TIF51A* (**b**), *eIF2A* (**c**), *SDH1* (**d**), *HAP4* (**e**) and *HAP1* (**f**) mRNA levels were determined. (**g**,**h**). A representative Western blotting experiment (**g**) and quantification analysis (**h**) of the eIF5A and eIF2A protein levels in the WT cells at the indicated time points. G6PDH protein levels were used as loading controls. The results are shown as the means ± SD of three independent experiments and are expressed in relation to the value at time 0. Statistical significance was measured by a Student’s *t*-test in relation to time 0. * *p* < 0.05, ** *p* < 0.01, *** *p* < 0.001.

**Figure 4 ijms-22-00219-f004:**
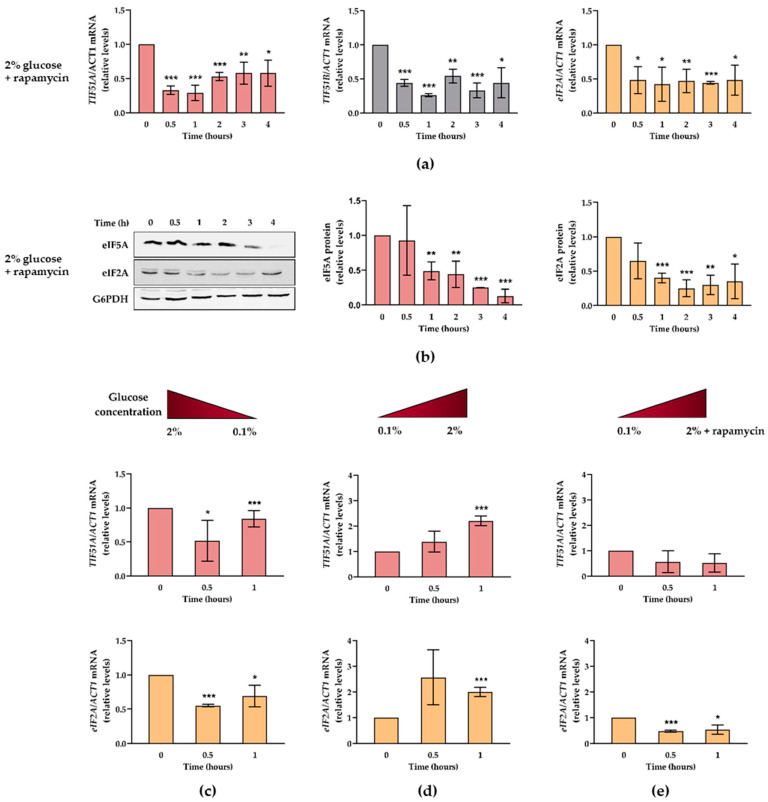
eIF5A expression is regulated by glucose concentration and TORC1 pathway. (**a**,**b**) WT cells were grown in YPD medium with the addition of 200 ng/mL rapamycin for 4 h. (**a**) The relative *TIF51A*, *TIF51B*, and *eIF2A* (from left to right) mRNA levels were determined. (**b**) Western blotting and quantification analysis of proteins eIF5A and eIF2A (from left to right). G6PDH protein levels were used as loading controls. A representative experiment is shown of three independent experiments. (**c**) The WT cells were grown in the YEP medium containing 2% glucose and transferred to the YEP medium containing 0.1% glucose for 1 h. (**d**,**e**) The WT cells were grown in the YEP medium containing 0.1% glucose and transferred to the YEP medium containing 2% glucose without (**d**) or with (**e**) the addition of rapamycin for 1 h. (**c**–**e**) The relative *TIF51A* (up) and *eIF2A* (down) mRNA levels were determined. (**a**–**e**) The results are shown as the means ± SD of three independent experiments and are expressed in relation to the value for time 0. Statistical significance was measured by a Student’s *t*-test in relation to time 0. * *p* < 0.05, ** *p* < 0.01, *** *p* < 0.001.

**Figure 5 ijms-22-00219-f005:**
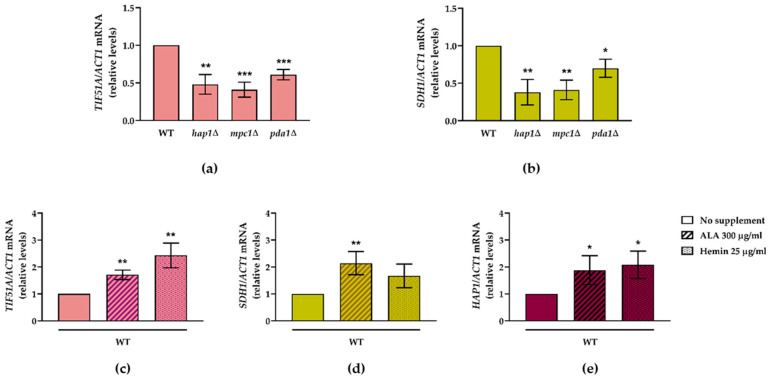
eIF5A expression is regulated by the metabolic flux into the TCA cycle and heme cellular levels. (**a**,**b**) The WT, *hap1*∆, *mpc1*∆ and *pda1*∆ strains were grown in YPGal medium for 24 h. The relative *TIF51A* (**a**) and *SDH1* (**b**) mRNA levels were determined. The results are expressed in relation to the WT value. (**c**–**e**) The WT cells were grown in YPD medium with or without the addition of ALA (300 µg/mL) or hemin (25 µg/mL) for 24 h. The relative *TIF51A* (**c**), *SDH1* (**d**) and *HAP1* (**e**) mRNA levels were determined. The results are expressed in relation to the value for the no supplement condition. (**a**–**e**) and are shown as the means ± SD of three independent experiments. Statistical significance was measured by a Student’s *t*-test. * *p* < 0.05, ** *p* < 0.01, *** *p* < 0.001.

**Figure 6 ijms-22-00219-f006:**
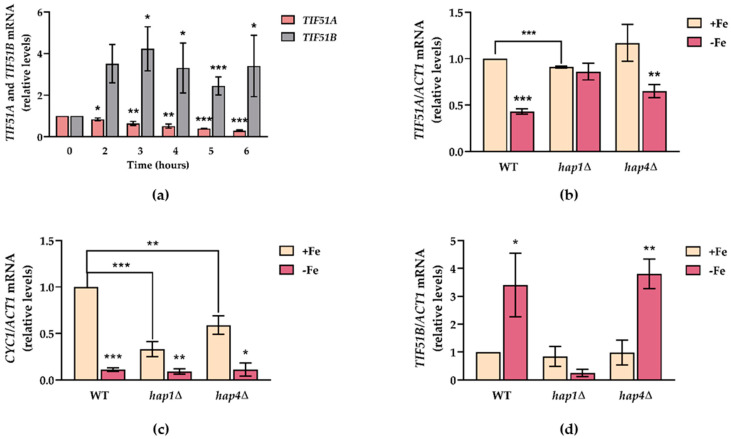
Iron deficiency down-regulates *TIF51A* and up-regulates *TIF51B* in a Hap1-dependent manner. (**a**) The WT cells were grown in SC medium with the addition of 100 mM BPS and samples were collected at the indicated time points. The relative *TIF51A* and *TIF51B* mRNA levels were determined. The results are expressed in relation to the value at time 0. (**b**–**d**) The WT, *hap1*∆ and *hap4*∆ strains were grown in SC medium with or without the addition of 100 mM BPS for 7 h. The relative *TIF51A* (**b**), *CYC1* (**c**) and *TIF51B* (**d**) mRNA levels were determined. The results are expressed in relation to the value of the WT strain without treatment. (**a**–**d**) The results are shown as the means ± SD of three independent experiments. Statistical significance was measured by a Student’s *t*-test. * *p* < 0.05, ** *p* < 0.01, *** *p* < 0.001.

**Figure 7 ijms-22-00219-f007:**
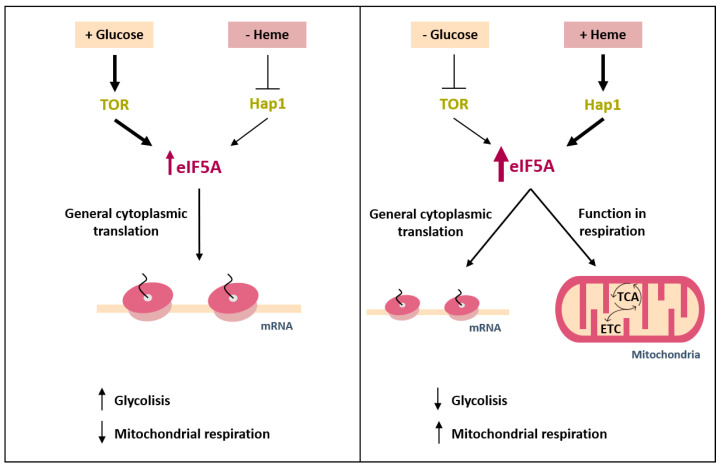
Model of regulation of the translation elongation factor eIF5A expression by nutrient availability through different signalling pathways. Under nutrient availability conditions, glucose signalling via TOR up-regulates eIF5A expression to facilitate cytoplasmic translation, and glycolysis predominates over mitochondrial respiration. Under low glucose and/or non-fermentative carbon sources, TOR positive eIF5A regulation is inhibited, but eIF5A expression is up-regulated by the increase in heme levels and subsequent Hap1 activation. In these conditions, eIF5A promotes, additionally to cytoplasmic translation, not-well known functions in mitochondrial respiration.

## Data Availability

Data is contained within the article or supplementary material. The data presented in this study are available in [insert article or supplementary material here].

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
