# Peer review of "Yeast Translation Elongation Factor eIF5A Expression Is Regulated by Nutrient Availability through Different Signalling Pathways"

_ijms, 2020, doi:10.3390/ijms22010219_

Round 1

Reviewer 1 Report

In this manuscript, Barba-Aliaga et. al analyzed the gene expression profiles of yeast Tif51A and Tif51B at different metabolic conditions, and they found Tif51A is essential for normal growth under non-fermentation conditions, while Tif51B was not observed to regulate mitochondrial respiration. The differential regulation of these two homologs genes suggests subfunctionalization and is interesting to the field. Please find the specific comments below.

  1. I would like to see the tone in the abstract and main results to be more confirmative. Please avoiding the wording “seemed to” or “likely”.
  2. Please consider adding another figure or supplemental figures to summarize the findings (signaling pathways).
  3. In the figures, the lower boundary of the error bar was covered by the barplot. Could the authors bring the lower error bar to the front?
  4. Line 379 Figure 6, is the upregulation of Tif51B to compensate the down-regulation of Tif51A due to iron depletion? Could the authors also discuss the normal function of TIF51B?

Author Response

Answers to the comments of the Reviewers and the modifications made in the manuscript according to the Reviewers’ opinions:

REVIEWER 1

Comments and Suggestions for Authors

In this manuscript, Barba-Aliaga et. al analyzed the gene expression profiles of yeast Tif51A and Tif51B at different metabolic conditions, and they found Tif51A is essential for normal growth under non-fermentation conditions, while Tif51B was not observed to regulate mitochondrial respiration. The differential regulation of these two homologs genes suggests subfunctionalization and is interesting to the field. Please find the specific comments below.

  1. I would like to see the tone in the abstract and main results to be more confirmative. Please avoiding the wording “seemed to” or “likely”.

Answer: Following Reviewer recommendation we changed in Abstract (new line 23) the sentence “Tif51A expression seemed to follow dual positive regulation” by the more confirmative “Tif51A expression followed dual positive regulation”. In Results (new line 381-2) we changed the sentence “Finally, Hap1 also seems important for TIF51B regulation…” by the more confirmative “Finally, Hap1 was also important for TIF51B regulation…”.

  1. Please consider adding another figure or supplemental figures to summarize the findings (signaling pathways).

Answer: It has been added new Figure 7 with a Model summarizing results and showing signaling pathways regulating eIF5A expression. It is cited in lines 423-24.

  1. In the figures, the lower boundary of the error bar was covered by the barplot. Could the authors bring the lower error bar to the front?

Answer: Lower boundary of error bars are now visible in all Figures as suggested.

  1. Line 379 Figure 6, is the upregulation of Tif51B to compensate the down-regulation of Tif51A due to iron depletion? Could the authors also discuss the normal function of TIF51B?

Answer: Upregulation of TIF51B gene does not compensate the downregulation of TIF51A. This is now shown in new supplementary Figure S2, and it is described in lines 364-366.

With respect to the second suggestion. We think that Tif51B function has been sufficiently discussed along the manuscript. First, in the Introduction (lines 74-80) it was described that Tif51A and Tif51B isoforms seem to have a common molecular role because both complement the growth of a TIF51A deleted yeast strain. But it is also pointed the possibility of a different function because both paralogue genes are differentially regulated, as it is shown along the manuscript.  Later, in the Discussion section (line 451 and so on) it was discussed: “It is tempting to suggest that the differential expression of the eIF5A isoforms would promote, in each case, differential metabolic outcomes with Tif51A promoting respiration and Tif51B promoting aerobic glycolysis.”  

Therefore, we believe that the function of Tif51B, what it is known and the hypothesis of a putative specific role in aerobic glycolysis, is discussed in the current version of the manuscript.

Reviewer 2 Report

Review

This manuscript focuses on different factors that influence the transcription of an elongation factor. eIF5 is a well-conserved elongation factor that is encoded by two paralogous genes (TIF51A and B) which are subject to different expression patterns. While TIF51A expression is induced in nonfermentable carbon sources that require respiration and iron starvation, TIF51B expression is repressed under these growth conditions. Knockouts of TIF51B don’t change the growth on different carbon sources. The authors then focus on the factors that regulate TIF51A expression. The tif51a mutant is lethal while the tif51b mutant doesn’t have a phenotype under conditions tested here and not further discussed. Both are hypusinated but the loss of the PTM doesn’t change oxygen consumption. While TIF51A transcription is positively regulated by Hap1 and Snf1, Msn2 negatively regulates expression in a similar pattern to SDH1, a nuclear-encoded mitochondrial protein. Expression of these genes was also measured as wildtype and hap1 mutant cells transited through the diauxic shift and entered stationary phase. Like growth in nonfermentable carbon, eIF5A increases as cells rely on respiration which coincides with the expression of HAP1. It appears that either paralog of eIF5 doesn’t tolerate epitope tags and the antibody recognizes both paralogs. So protein expression wasn’t measured. With the requirement of Tif51a in growth on non-fermentable carbon sources, the expression level was measured in cells grown on galactose media supplemented with either heme or a precursor, and expression of TIF51A increased. Despite increased eIF5A expression in the post-diauxic shift at 24 hours protein disappears and then reappears at 96 hours. A slightly different regulation pattern was found when cells were treated with rapamycin and expression decreased within 30 minutes and protein levels decrease shortly after and at 4 hours nearly are gone.

Figure 1d The tif51A-3 mutant which is more severe runs slower than the other mutant at 25 degrees. Could that be from a change in the overall charge of the protein or a PTM? How much do you estimate the shift is in kDa?

Figure 2a should be done as serial dilutions as in Figure 1a to better determine differences in growth between the mutants on media with ethanol.

How does the protein levels of SDH1 change in the tif51a mutants from Figure 1? The authors posit that because Tif51a has been found associated with mitochondria that it is important for the expression of nuclear-encoded mitochondrial genes.

Do these mitochondrial transcripts measured here or in general share any motifs in common? How long are their 5' leader sequence? Are there motifs or uORFs in the 5' leader sequence? Are there stretches of homopolymers such as prolines, glycines, or other unusual protein sequences? How does the distribution of codon bias look through the coding region? That can be calculated on codons.org (Clarke and Clark 2008).

Are the tif51a mutants from Figure 1 sensitive when grown on media with BPS?

Figure 6 legend needs to be clearer. Were cells grown in SC and then treated with BPS? Or treated for 6 hours and then collected over the next six hours?

In the Discussion, lines 415-426, the authors discuss how it is unknown how eIF5A controls the expression of mitochondrial-targeted mRNAs. The authors stop short of showing that the translation of transcripts such as SDH1 is regulated by eIF5A. A western blot of tif51a from figure 1 should show that when grown in non-fermentable media that Sdh1 protein levels decrease.

There seems to be a disconnect between what the authors measure and the function of eIF5A. eIF5A associates with ribosomes. Literature suggests that eIF5A has a role in elongation and is found associated with the mitochondria. However, most of the studies here are on factors that regulate TIF51A or nuclear genes that encode mitochondrial destined proteins. There are many instances where increased transcription doesn’t correlate with increased translation. In particular Figure 3b and g show that is the case for Tif51a. Protein levels of Sdh1 and Cyc1 should be measured in the tif51a mutants in nonfermentable carbon sources.

Line 422 SDH1 should be in italics.

Author Response

REVIEWER 2

Comments and Suggestions for Authors

This manuscript focuses on different factors that influence the transcription of an elongation factor. eIF5 is a well-conserved elongation factor that is encoded by two paralogous genes (TIF51A and B) which are subject to different expression patterns. While TIF51A expression is induced in nonfermentable carbon sources that require respiration and iron starvation, TIF51B expression is repressed under these growth conditions. Knockouts of TIF51B don’t change the growth on different carbon sources. The authors then focus on the factors that regulate TIF51A expression. The tif51a mutant is lethal while the tif51b mutant doesn’t have a phenotype under conditions tested here and not further discussed. Both are hypusinated but the loss of the PTM doesn’t change oxygen consumption. While TIF51A transcription is positively regulated by Hap1 and Snf1, Msn2 negatively regulates expression in a similar pattern to SDH1, a nuclear-encoded mitochondrial protein. Expression of these genes was also measured as wildtype and hap1 mutant cells transited through the diauxic shift and entered stationary phase. Like growth in nonfermentable carbon, eIF5A increases as cells rely on respiration which coincides with the expression of HAP1. It appears that either paralog of eIF5 doesn’t tolerate epitope tags and the antibody recognizes both paralogs. So protein expression wasn’t measured. With the requirement of Tif51a in growth on non-fermentable carbon sources, the expression level was measured in cells grown on galactose media supplemented with either heme or a precursor, and expression of TIF51A increased. Despite increased eIF5A expression in the post-diauxic shift at 24 hours protein disappears and then reappears at 96 hours. A slightly different regulation pattern was found when cells were treated with rapamycin and expression decreased within 30 minutes and protein levels decrease shortly after and at 4 hours nearly are gone.

  1. Figure 1d The tif51A-3mutant which is more severe runs slower than the other mutant at 25 degrees. Could that be from a change in the overall charge of the protein or a PTM? How much do you estimate the shift is in kDa?

Answer: The mutations in eIF5A present in the tif51A-1 (P83S) and tif51A-3 (C39Y and G118D) strains were obtained by the group of Sandro Valentini (Valentini et al., (2002) Genetics, 160, 393-405). In their work they described that the tif51A-3 protein migrates slightly more slowly than the wild-type Tif51A and tif51A-1 mutants due to the mutation of glycine 118 to aspartic present in eIF5A. This explanation has been now included in Figure 1 legend for clarity (lines 193-194).

  1. Figure 2a should be done as serial dilutions as in Figure 1a to better determine differences in growth between the mutants on media with ethanol.

Answer: Although we agree with the Reviewer that differences in growth are more precisely shown by plating serial dilutions of cells, in this case we believe that it is clearly shown the inhibition of growth in hap4 mutant, the slower growth of snf1 mutant and the normal growth of hap1 mutant in media with ethanol. Moreover, a representative plate is shown out of several repetitions. Therefore, we think it is not necessary to reconfirm these results with serial dilution plating.

  1. How does the protein levels of SDH1 change in the tif51a mutants from Figure 1? The authors posit that because Tif51a has been found associated with mitochondria that it is important for the expression of nuclear-encoded mitochondrial genes.

Answer: This question of Reviewer 2 and some of his/her next questions revolve around the function of eIF5A in mitochondrial respiration. Of course, this is a main goal to be pursued, however, it is not the goal of the present work, that it is to study the regulation of eIF5A expression, both isoforms, in different metabolic conditions. Therefore, we think that the proposed experiment is out of the objectives of this work.

  1. Do these mitochondrial transcripts measured here or in general share any motifs in common? How long are their 5' leader sequence? Are there motifs or uORFs in the 5' leader sequence? Are there stretches of homopolymers such as prolines, glycines, or other unusual protein sequences? How does the distribution of codon bias look through the coding region? That can be calculated on codons.org (Clarke and Clark 2008).

Answer: Again the Reviewer questions are around eIF5A control of mitochondrial genes, and, as explained in answer to question 3, this is not the scope of this work. In any case, our results suggest a function of eIF5A in mitochondrial respiration, and this is discussed along the manuscript. Moreover, previous results from Puleston et al (ref. 34 in the manuscript) focused in this direction and showed that depletion of functional eIF5A results in lower protein levels of some TCA enzymes, suggesting a control by eIF5A of their translation efficiency. However, the mechanism of control is not clear, as it is discussed in Puleston et al., and it is not understood the specificity of this mechanism, affecting some TCA enzymes, but not all. Therefore, it is not an obvious or easy matter to figure out how eIF5A promotes respiration. The simplest hypothesis would be that eIF5A is required for the translation of TCA or OXPHO proteins because some of these proteins may contain the described eIF5A-dependent motifs. In this direction we have now included, in the revised version of the manuscript, a supplementary Figure S3 showing the TCA enzymes and OXPHO proteins content in the described-eIF5A tripeptide motifs (Pelechano and Alepuz, 2017, ref.2 in the manuscript) and these results are introduced now in the Discussion section, lines 428-432. As a consequence, several slight changes in this paragraph have been made.  

  1. Are the tif51a mutants from Figure 1 sensitive when grown on media with BPS?

Answer: We have checked this out and found that TIF51A temperature sensitive mutants at restrictive temperature have growth defects under iron starvation. We have included this result in the revised version of the manuscript and we have interpreted it as described now in lines 366-369:” The increased but low expression of the Tif51B isoform must be insufficient to support yeast growth because we observed that complete depletion of Tif51A protein (using TIF51A temperature sensitive mutants at restrictive temperature) rendered sensitivity to iron starvation (Suppl. Figure S3).”

  1. Figure 6 legend needs to be clearer. Were cells grown in SC and then treated with BPS? Or treated for 6 hours and then collected over the next six hours?

Answer: We thank Reviewer 2 for noting this. It was a mistake in the Figure legend that it is now corrected. So, cells were grown in SC media and then treated BPS and collected at the indicated times. Accordingly, Figure 6 legend has been corrected.

  1. In the Discussion, lines 415-426, the authors discuss how it is unknown how eIF5A controls the expression of mitochondrial-targeted mRNAs. The authors stop short of showing that the translation of transcripts such as SDH1 is regulated by eIF5A. A western blot of tif51afrom figure 1 should show that when grown in non-fermentable media that Sdh1 protein levels decrease.

Answer: See answer to questions 3 and 4.

  1. There seems to be a disconnect between what the authors measure and the function of eIF5A. eIF5A associates with ribosomes. Literature suggests that eIF5A has a role in elongation and is found associated with the mitochondria. However, most of the studies here are on factors that regulate TIF51A or nuclear genes that encode mitochondrial destined proteins. There are many instances where increased transcription doesn’t correlate with increased translation. In particular Figure 3b and g show that is the case for Tif51a. Protein levels of Sdh1 and Cyc1 should be measured in the tif51a mutants in nonfermentable carbon sources.

Answer: See answer to questions 3 and 4.